# Rat Sympathetic Neuron Calcium Channels Are Insensitive to Gabapentin

**DOI:** 10.3390/ph17091237

**Published:** 2024-09-19

**Authors:** Mallory B. Scott, Paul J. Kammermeier

**Affiliations:** Department of Pharmacology and Physiology, University of Rochester Medical Center, Rochester, NY 14642, USA

**Keywords:** voltage-gated calcium channel, patch clamp, channel modulation, gabapentin, electrophysiology, trafficking

## Abstract

The gabapentenoids such as gabapentin (GP) and pregabalin are approved for the treatment of chronic pain, but their utility is limited by persistent side effects. These adverse effects result from GPs affecting many types of neurons and muscle cells, not just the pain-sensing neurons that are the intended targets. We have recently discovered a type of peripheral neuron, rat sympathetic neurons from the superior cervical ganglion (SCG), that is uniquely insensitive to GP effects. Currents were measured using whole-cell patch-clamp electrophysiology from cells in primary culture from either the SCG or the Nodose Ganglion (NDG) as a positive control for the effects of GP. We find that the calcium current density was dramatically reduced by GP pretreatment in NDG neurons, but that neurons from the SCG were resistant. Further, when GP was cytoplasmically injected into these neurons, the resistance of SCG neurons to GP treatment persisted. These data demonstrate that rat sympathetic neurons appear to be uniquely resistant to GP treatment. These results may help us to better understand the mechanism of action of, and resistance to, GP in altering calcium channel current density, which may help to develop future treatments with fewer side effects.

## 1. Introduction

Gabapentinoids (GPs) have been an effective and important tool in the treatment of pain disorders, but their widespread use has been hindered by numerous adverse side effects [1,2]. GPs bind to CaV α2δ subunits and reduce the number of voltage-gated CaV channels on the plasma membrane of excitable cells [3]. Since these channels are responsible for permitting calcium entry that leads to neurotransmitter release, the likely mechanism of action of GPs is to reduce neurotransmitter release from pain-sensing neurons [4], leading to a reduced sensation of pain, although other actions of GP may play a role [5]. However, CaV channels that require α2δ subunits are expressed in nearly every type of neuron and in other excitable cells such as skeletal muscle. Therefore, many of the adverse effects of GP treatment may be due to a reduction in CaV channels in muscle and off-target neurons [6,7,8,9]. To our knowledge there have been no specific neuronal subtypes reported that show resistance to GPs. A better understanding of the unique properties of SCG neurons that impart GP resistance will be extremely valuable and could lead to the development of more effective and more targeted treatment of chronic pain disorders.

Voltage-gated calcium channels, or CaV channels, mediate several important functions by coupling cell depolarization to processes such as muscle contraction (EC coupling), neurotransmitter and hormone secretion (ES coupling), and long-term changes in gene transcription (ET coupling), often through activation of phospho-CREB [10,11]. There are three classes of Cav channel (Cav1-3). The Cav1 and Cav2 families are activated at higher voltages and require accessory subunit association. The Cav3 channels, or “T-type” channels, comprise the low-voltage-activated channels characterized by their activation at relatively negative potentials and their strong, complete inactivation at depolarized potentials. CaV3 channels do not require accessory subunits and as such are not modulated by GPs.

While the pore-forming, α1 subunits of calcium channels contain all of the structural and functional motifs that underlie basic channel function (voltage sensing, permeation, gating, etc.), Cav1 and Cav2 channels also require association with accessory subunits β and α2δ. A recently published structure of the channel complex with CaV1.1, α2δ, and β subunits has begun to shed light on the physical interactions of these proteins [12]. The functional roles of accessory subunits are complex but center around their ability to promote protein expression, traffic new protein to the plasma membrane, and stabilize the channels once they are there [13,14]. However, it is not clear whether all of these subunits are tightly associated at all times on the plasma membrane, though α2δ appears to be [15]. Co-expression of β subunits with Cav1 and Cav2 invariably greatly enhances channel expression (and currents) [16] and alters the biophysical properties of the channels [14].

The α2δ subunits are multi-domain subunits covalently linked but translated from a single gene [17]. α2δ is mainly an extracellular protein with a GPI anchor, although deletion of the anchor does not impair its ability to regulate CaV channels [18]. Co-expression of any α2δ subunit with a β subunit and a CaV1 or CaV2 channel (in HEK cells, for example) further enhances the current density [14]. The mechanism for this increased current density is often attributed to general “trafficking” effects, but the precise role of α2δ in calcium channel trafficking is complex. A recent study appears to have uncovered a role for α2δ, primarily in the context of the action of GPs on calcium currents [19]. In that study, the authors found that the effect of GPs, which have been widely reported to bind to α2δ-1 or -2 to ultimately reduce calcium current density in a variety of cell types [8,9], was to prevent one specific function of α2δ, namely the recycling of Cav channel complexes, previously internalized, back to the plasma membrane. The model in Figure 1 illustrates a simplified but useful model describing these roles, wherein the primary role of β subunits is to facilitate newly translated Cav channels from the ER, through the secretory pathway to the plasma membrane (★), while the association with α2δ subunits reduces the rate of internalization, and in the endosomal compartment, it promotes recycling of channels previously internalized into endosomes back to the plasma membrane (★) [19]. This model is consistent with much of what is known about the role of accessory subunits on CaV channel expression and trafficking. Assuming some constitutive rate of new protein synthesis as well as channel internalization followed by either degradation or recycling, it can explain how the expression of both β and α2δ expression can enhance current density.

In this study, the voltage calcium current density was examined in two rat peripheral neuronal cell types: sympathetic neurons from the superior cervical ganglion (SCG) and mixed visceral sensory neurons from the Nodose Ganglion (NDG). While NDG neurons exhibited marked reductions in calcium current density upon application of GP, a surprising resistance to GP was seen with SCG neurons.

## 2. Results

In following up on a recent study [20] examining the role of α2δ subunits in regulating Cav channel expression, we were surprised to find that gabapentin failed to reduce calcium current density in rat SCG neurons, despite expressing α2δ-1 and -2, and Cav2.2 and 2.3. This finding prompted us to perform some more careful experiments. To that end, neurons were isolated from both the SCG and the NDG from adult rats and were kept in primary culture overnight. Indeed, we found that calcium current density in NDG neurons was dramatically reduced following overnight 1 mM GP treatment as expected, while currents in SCG neurons were not detectably changed (Figure 2). IV curves were generated by stepping from a holding potential of −80 mV to a range of test voltages from −80 to +80 mV for 80 msec. Figure 2A,B shows a few sample current traces from this range, as indicated (*upper*) from an untreated control cell (*center*) and a cell treated with 1 mM GP applied to the media overnight (*lower*). The range of current traces shown was chosen to show nearly the full range of voltage steps that elicited current, with the two most negative steps shown in gray to distinguish from the currents elicited at more positive voltages. Capacitance transients from the depolarizing step are deleted for clarity and aesthetic purposes, but tail currents are not truncated.

Figure 2C,D show average IV curves from 10 to 11 cells (as indicated; average ± SEM), from SCG neurons (Figure 2C) and NDG neurons (Figure 2D), and from control cells (filled symbols) and those pretreated with 1 mM GP (open symbols). Current measurements were made as the average of all points between 9 and 11 msec after the start of each voltage step. As indicated, none of the currents elicited at any voltages in SCG neurons were statistically distinguishable in the control vs. GP-treated cells, but currents from NDG neurons were significantly smaller in GP-treated cells compared with control cells when elicited at voltages from −10 mV to +30 mV, as indicated (*; *p* ≤ 0.05, Student’s *t*-test performed using current densities at each voltage). Together these data indicate that, as expected, voltage-gated calcium current density is reduced with GP pretreatment in NDG neurons, as has been demonstrated with several other excitable cell types [6,7,8,9]. Surprisingly, however, neurons from the SCG were uniquely resistant to GP treatment.

Next, because GPs require transport into cells [21], it is possible that the resistance of SCG calcium channels to GP may be due to the absence of the necessary transporters. Thus, we asked whether cytoplasmically injected GP could reduce the calcium currents in NDG and SCG neurons. To accomplish this, cultured neurons were injected both cytoplasmically and nuclearly with a solution containing either just 200 ng/µL of EGFP cDNA (Figure 3 Con; filled symbols) or EGFP plus 1 mM GP (Figure 3 GP; open symbols). Patch-clamp recordings were made the following day from cells showing GFP fluorescence for both groups to control for any potential effects of injection. Interestingly, injected GP was as effective as bath-applied GPs at reducing NDG calcium currents (Figure 3, *right*) but remained ineffective at reducing the currents in SCG neurons (Figure 3, *left*). These data establish SCG neurons as a surprisingly GP-resistant neuronal subtype.

## 3. Discussion

In the current study, the ability of GP to modulate calcium currents from rat sympathetic neurons from the SCG was examined. Because SCG neurons express only high-voltage-activated CaV2 channels that require association with accessory subunits including α2δ, we expected that GP treatment would result in current density reduction as it does in many other documented neural and other excitable cell types [6,7,8,9]. Interestingly, a reduction in current density was not observed. As a positive control, we also examined NDG neurons from rats and found that these were indeed significantly reduced by similar GP treatment, suggesting that SCG neurons may be surprisingly resistant to GP.

One documented explanation for the lack of effect of GP on calcium currents is that GP may fail to be taken up by these cells. Although GP binds to an exofacial site on the α2δ subunit, it nevertheless does appear to require transport into cells for activity on calcium channels and ultimately neurotransmitter release [21]. Thus, we tested whether the lack of effect on SCG neurons may have been due to GP’s inability to gain access to intracellular compartments. To accomplish this, we simply injected GP cytoplasmically instead of extracellularly into the media. We were surprised to find that cytoplasmic injection of GP was sufficient to reduce NDG calcium current density, but we still observed resistance in SCG neurons, suggesting that these neurons may be uniquely resistant to modulation by GP for reasons yet unknown.

While SCG and NDG neurons exhibit some differences in their calcium channel expression, including larger overall current densities in NDG [22], both express primarily Cav1 and 2 channels that are expected to be modulated by GPs. Our data suggest that a more thorough comparison of these cell types may be warranted to better understand SCG GP resistance. Obvious hypotheses include that SCG neurons express some factor or factors that render them resistant to GP, or that they lack a protein needed for GP actions on calcium channels. Indeed, a recently published transcriptomic analysis comparing rat SCG, NDG, and other neuronal species such as dorsal root and trigeminal ganglia [23] may provide a starting point to begin to address these kinds of questions.

These findings provide an important example of a neuron that shows resistance to GP treatment, which may ultimately lead to a better understanding of GP’s actions. In time, a better understanding of the mechanism of SCG neuron resistance to GP may help lead to the development of more selective therapeutics that can modulate channels in pain sensory neurons without dramatically altering those in other excitable cell types, potentially reducing unwanted side effects.

## 4. Materials and Methods

### 4.1. Cell Isolation and Culture

Neuron isolation protocols have been described previously [20,24]. SCG and NDG were isolated from adult male Wistar rats. Briefly, both of the ganglia are located near the major bifurcation of the carotid artery located in the neck of the rat, with the SCG sitting just at the bifurcation and the NDG slightly more caudally along the cervical carotid triangle. Each ganglion was dissected out, removed from a connective tissue sheath, and then incubated in Earle’s balanced salt solution (EBSS) (Life Technologies, Rochelle, MD, USA) with 0.5–0.6 mg/mL trypsin (Worthington, Freehold, NJ, USA) and 1.6 mg/mL Type IV collagenase (Worthington) for 1 h at 35 °C. Cells were pelleted, transferred to minimum essential medium (MEM; Gibco/Thermo-Fisher, Waltham, MA, USA), plated, and incubated at 37 °C until recording the following day. Injections were performed (when indicated) with an Eppendorf 5247 microinjector and an Injectman NI2 micromanipulator (Madison, WI, USA) for 4–6 h following cell isolation. pEGFP plasmids (from Clontech, now Takara Biosciences, Kusatsu, Shiga, Japan) were stored at −20 °C as a 1–2 μg/μL stock solution in TE buffer (10 mM TRIS, 1 mM EDTA, and pH 8). Concentrations of cDNAs injected were as indicated in the text. Cells were incubated overnight at 37 °C, and experiments were performed the following day. All animal protocols were approved by the University of Rochester’s Committee on Animal Resources (UCAR).

### 4.2. Electrophysiology

Patch-clamp recordings were made using 8250 glass (King Precision Glass, Claremont, CA, USA). Pipette resistances were 1–3 MΩ leading to uncompensated series resistances of 1–5 MΩ. Series resistance compensation of 80% was used in all recordings. Data were recorded using an Axon Axopatch 200B patch-clamp amplifier (Molecular Devices, Sunnyvale, CA, USA). Voltage protocol generation and data acquisition were performed using custom procedures written within the Igor Pro 9 software package (Wavemetrics, Lake Oswego, OR, USA) by Stephen R. Ikeda (NIH, NIAAA) on a MacMini Intel DuoCore computer with an Instrutech ITC18 data acquisition board (HEKA Elektronik/Harvard Bioscience, Holliston, MA, USA). Currents were sampled at 100 kHz, low-pass filtered at 5 kHz, digitized, and stored for analysis. All experiments were performed at 21–24 °C. Data analysis was performed using Igor Pro software (WaveMetrics, Lake Oswego, OR, USA). To isolate calcium currents, Na and K were replaced with large, impermeant cations yielding external recording solutions containing (in mM) 145 tetraethylammonium (TEA) methanesulfonate (MS), 10 4-(2 Hydroxyethyl)-1-piperazineethanesulfonic acid (HEPES), 15 glucose, 10 CaCl_2_, and 300 nM tetrodotoxin, with a pH of 7.4 and an osmolality of 320 mOsm/kg, and an internal (pipette) solution containing 120 N-methyl-D glucamine (NMG) MS, 20 TEA, 11 EGTA, 10 HEPES, 10 sucrose, 1 CaCl_2_, 4 MgATP, 0.3 Na2GTP, and 14 tris- creatine phosphate, with a pH of 7.2 and an osmolality of 300 mOsm/kg. GP was obtained from Millipore-Sigma (Burlington, MA, USA).

## Figures and Tables

**Figure 1 pharmaceuticals-17-01237-f001:**
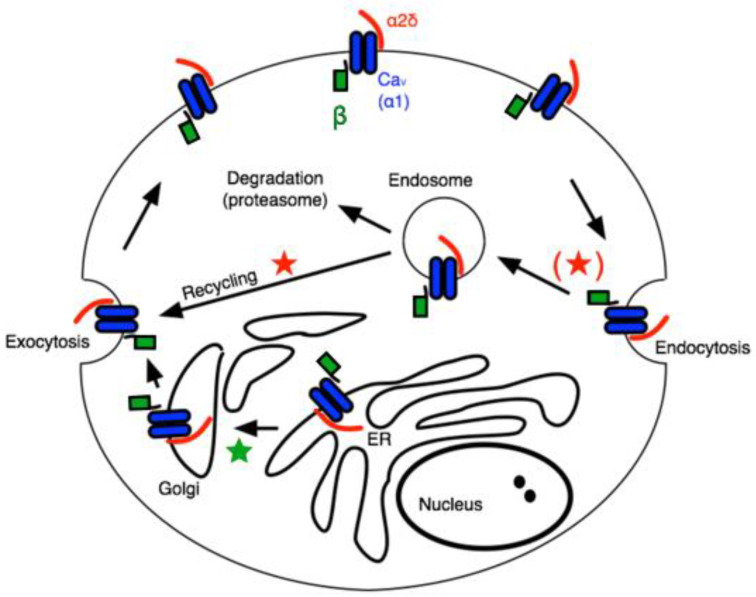
Schematic of calcium channel trafficking and points of potential impact of auxiliary subunits. Pore-forming ⍺1 subunits are shown in blue, β subunits are shown in green, and α2δ subunits are red. Stars indicate potential points of impact for β (★) and α2δ subunits (★).

**Figure 2 pharmaceuticals-17-01237-f002:**
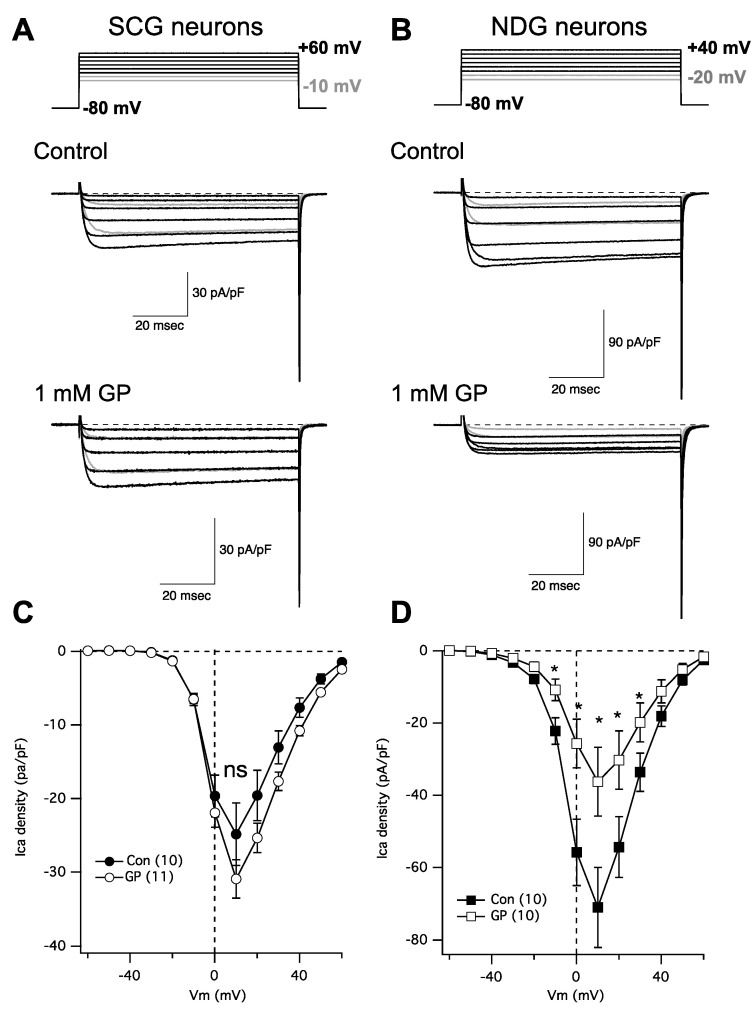
GP treatment reduces calcium current density in NDG neurons but not SCG neurons. (**A**) Currents from SCG neurons; *upper*: voltage protocol for the portion of the IV curve illustrated *below* for untreated (control) and GP-treated neurons (1 mM GP) from example cells. (**B**) Currents from NDG neurons; *upper*: voltage protocol for the portion of the IV curve illustrated *below* for untreated (control) and GP-treated neurons (1 mM GP) from example cells. (**C**) Average current densities from a full IV curve applied to SCG neurons (number of cells indicated in parentheses) in control and GP-treated cells. (**D**) Average current densities from a full IV curve applied to NDG neurons (number of cells indicated in parentheses) in control and GP-treated cells. * indicates a significantly different current density compared with control cells from the same voltage, *p* ≤ 0.05, *t*-test.

**Figure 3 pharmaceuticals-17-01237-f003:**
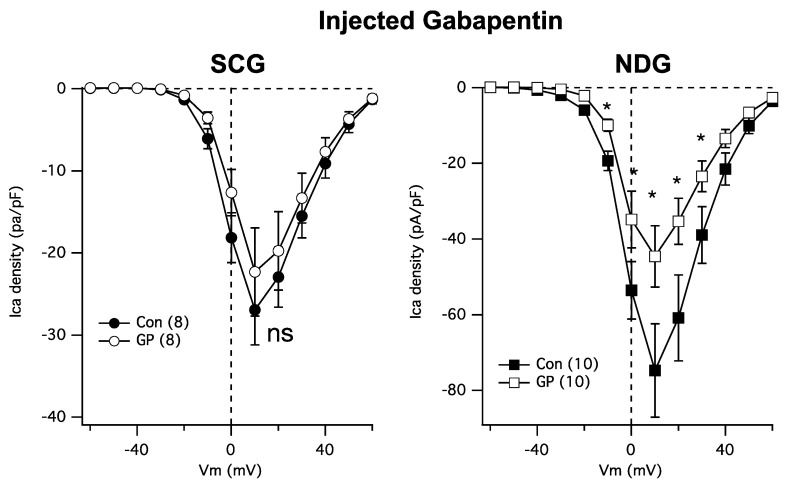
SCG neuron calcium currents are similarly resistant to GP treatment when intracellularly injected. ***Left***: average current densities from a full IV curve applied to SCG neurons (number of cells indicated in parentheses) in control and GP (1 mM)-injected cells. ***Right***: average current densities from a full IV curve applied to NDG neurons (number of cells indicated in parentheses) in control and GP-injected cells. * indicates a significantly different current density compared with control cells from the same voltage, *p* ≤ 0.05, *t*-test.

## Data Availability

The original contributions presented in the study are included in the article, further inquiries can be directed to the corresponding author.

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
