# Peer review of "Rat Sympathetic Neuron Calcium Channels Are Insensitive to Gabapentin"

_pharmaceuticals, 2024, doi:10.3390/ph17091237_

Round 1

Reviewer 1 Report

Comments and Suggestions for Authors

Comments are listed here for the author’s consideration to further improve the quality and overall impact of the manuscript.

If authors focused in general on calcium channel current density, why is it important to mention all subtypes of calcium channels at introduction? Line 41-57. The same for the subunits of calcium channels (Line 59-71), it is not relevant information. Correct the Introduction.

Why authors using 10 mM of gabapentin as treatment of cells? It was 10 or 1 mM of gabapentin? Correct it.

Explain how and from where the neurons were isolated.

How did you isolate the calcium current in cell cultures? Explain it.

Why didn't you isolate specific calcium channel currents?

Line 117. “….sample current traces from a range of voltage steps…” Explain how you made your voltage protocol at methodology.

Why authors injected 200 ng/μl of EGFP cDNA to cells? How were the pEGFP plasmids obtained? Explain it at the methodology.

It lacks a discussion of the results obtained and their implications with the pathophysiology of pain and side effects of gabapentin associated with these peripheral neurons (sympathetic neurons from the superior cervical ganglion), taking into account that a major limitation is that they are results of cell cultures. What is the relevance of the study?

Comments on the Quality of English Language

English edition is mandatory, as there are some errors in the manuscript.

Author Response

If authors focused in general on calcium channel current density, why is it important to mention all subtypes of calcium channels at introduction? Line 41-57. The same for the subunits of calcium channels (Line 59-71), it is not relevant information. Correct the Introduction.

We include this in the introduction as background to explain that only the Cav1 and 2 channels require accessory subunits, and are therefore expected to be modulated by gabapentin. Further, an introduction as the the current conventional wisdom on how gabapentin acts at alpha2delta subunits to alter current density seems relevant in helping a reader interpret the data. 

Why authors using 10 mM of gabapentin as treatment of cells? It was 10 or 1 mM of gabapentin? Correct it.

“10 mM” was a typo and has been corrected. Treatment with gabapentin was done with 1 mM in all of the experiments.

Explain how and from where the neurons were isolated.

We have added the following text in the Materials and Methods section: “Briefly, both of the ganglia are located near the major bifurcation of the carotid artery located in the neck of the rat, with the SCG sitting just at the bifurcation and the NDG slightly more caudally along the cervical carotid triangle. Each ganglion was dissected out, removed from a connective tissue sheath, then incubated in Earle’s balanced salt solution… ”

How did you isolate the calcium current in cell cultures? Explain it.

This was done by replacing other permeant ions with impermeant ions. We have added this sentence in the Methods section: “To isolate calcium currents, Na and K were replaced with large, impermeant cations yielding external recording solution containing (in mM)

Why didn't you isolate specific calcium channel currents?

Because GP can alter the current density of any Cav1 and Cav2 channels, we found it better to not block any of these. We did not see evidence of Cav3 Chanels in SCG or NDG.

Line 117. “….sample current traces from a range of voltage steps…” Explain how you made your voltage protocol at methodology.

In each group, currents were elicited at a range of voltages from -80 to +80 mV, but sample current traces are shown from only a subset of this range. This is now more clearly explained in the Results section. 

Why authors injected 200 ng/μl of EGFP cDNA to cells? How were the pEGFP plasmids obtained? Explain it at the methodology.

pEGFP was from Clontech, which has been since absorbed by Takara Biosciences. We now attribute them in the Methods section.

It lacks a discussion of the results obtained and their implications with the pathophysiology of pain and side effects of gabapentin associated with these peripheral neurons (sympathetic neurons from the superior cervical ganglion), taking into account that a major limitation is that they are results of cell cultures. What is the relevance of the study?

We have added a short discussion that summarizes our study and provides some context as to the significance of our data. 

Reviewer 2 Report

Comments and Suggestions for Authors

The present manuscript by Scott & Kammermeier reports on rat sympathetic neurons from the Superior Cervical Ganglion (SCG) that are insensitive to gabapentin. The authors measured currents from primary cells using whole-cell patch clamp electrophysiology. Nodose Ganglion (NDG) neurons were thereby used as a positive control for the effects of gabapentin. These findings could be quite interesting for the development of future chronic pain treatments with fewer side effects compared to those utilizing gabapentin.

The manuscript is labeled as a “Brief Report” and therefore quite short, but it is nonetheless well-written and nicely structured. I still have a few remarks that the authors need to take care of:

·         Were the voltage protocols used for the SCG and NDG neurons different or the same? In Figure 2, the different labeling in panels A and B gives that impression.

·         NDG control neurons show much larger current densities (70 pA/pF) compared to SCG neurons (25 pA/pF). Is it possible that SCG neurons are simply missing some unknown molecular player that is also being blocked by gabapentin? Please discuss.

·         In conjunction with the previous comment, I found that the “Results and Discussion” section does not really discuss the findings appropriately. Please elaborate more on the results and/or add a separate discussion section.

·         Student’s T-test was used for statistical comparison. This test requires the data to be normally distributed as well as having homogeneous variance. Did the authors check for this? Also, a “Statistics” section should be added to “Materials and Methods”.

·         I recommend changing the color scheme used in Figure 1 from red/orange to some other combination that is easier to distinguish.

·         There are a few small textual issues remaining in lines 31/32, 121, 135 and 185.

Author Response

The present manuscript by Scott & Kammermeier reports on rat sympathetic neurons from the Superior Cervical Ganglion (SCG) that are insensitive to gabapentin. The authors measured currents from primary cells using whole-cell patch clamp electrophysiology. Nodose Ganglion (NDG) neurons were thereby used as a positive control for the effects of gabapentin. These findings could be quite interesting for the development of future chronic pain treatments with fewer side effects compared to those utilizing gabapentin.

The manuscript is labeled as a “Brief Report” and therefore quite short, but it is nonetheless well-written and nicely structured. I still have a few remarks that the authors need to take care of:

·         Were the voltage protocols used for the SCG and NDG neurons different or the same? In Figure 2, the different labeling in panels A and B gives that impression.

The same. We have amended the text with a more thorough description of the voltage protocols. 

·         NDG control neurons show much larger current densities (70 pA/pF) compared to SCG neurons (25 pA/pF). Is it possible that SCG neurons are simply missing some unknown molecular player that is also being blocked by gabapentin? Please discuss.

Clearly NDG neurons and SCG neurons are different in many ways. Indeed, the underlying reason for SCG neuron GP resistance is not known, but may include the possibility that SCG neurons are missing one or more factors that allow GP modulation, or that they express something that renders them resistant. We now discuss this briefly in the Discussion section.

·         In conjunction with the previous comment, I found that the “Results and Discussion” section does not really discuss the findings appropriately. Please elaborate more on the results and/or add a separate discussion section.

We have added a short Discussion section.

·         Student’s T-test was used for statistical comparison. This test requires the data to be normally distributed as well as having homogeneous variance. Did the authors check for this? Also, a “Statistics” section should be added to “Materials and Methods”.

Data from our 2017 paper suggest that SCG current densities are reasonably fit to a normal distribution (from about 1000 cells). We don’t have as much data on NDG neurons to make the same comparison so we do have to make the assumption that these are also normally distributed. Because the statistical tests run here are exceedingly simple, we did not feel that a separate section was warranted. 

·         I recommend changing the color scheme used in Figure 1 from red/orange to some other combination that is easier to distinguish.

We now use blue, red, and green instead of Blue, red, and orange. 

·         There are a few small textual issues remaining in lines 31/32, 121, 135 and 185.

We have amended these errors. 

Reviewer 3 Report

Comments and Suggestions for Authors

 The paper by Scott & Kammermeir reports very interesting findings about the sensitivity of Cav channels expressed in rat sympathetic neurons to gabapentin (GP), a widely used analgesic in pain therapy. The authors show very clearly that the calcium currents recorded from isolated rat Superior Cervical Ganglion (SCG) neurons are insensitive to GP compared to Nodose Ganglion (NDG) neurons in which GP reduces drastically the same type of calcium currents. The two figures of the paper reporting data are fine. The introductory figure has some problem with the colors which are not visibly very different and create confusion. Orange and Red do not appear very different in the printed figure.

I do not have major criticisms, but I suggest the authors to rewrite part of the Introduction, is over too long (many details are not necessary), and give more emphasis to the Discussion, which nearly does not exist.

 1 - In the Results & Discussion the first sentence: “In following up a recent study examining the role of α2δ subunits in regulating Cav channel expression, we were surprised to find that Gabapentin failed to reduce calcium current density in rat SCG neurons, despite expressing α2δ-1, & 2, and CaV2.2 & 2.3 (Scott & Kammermeir, 2017)” is not correct. The authors did not use and not even mention GP in their paper. The sentence should be rewritten.

2 - The authors do not discuss the various possibilities by which GP is inactive on the Cav channels of SCG compared to NDG Cav channels. Are the two types of Cav channels structurally similar? do they interact similarly with the a2d subunits? etc.

Author Response

The paper by Scott & Kammermeir reports very interesting findings about the sensitivity of Cav channels expressed in rat sympathetic neurons to gabapentin (GP), a widely used analgesic in pain therapy. The authors show very clearly that the calcium currents recorded from isolated rat Superior Cervical Ganglion (SCG) neurons are insensitive to GP compared to Nodose Ganglion (NDG) neurons in which GP reduces drastically the same type of calcium currents. The two figures of the paper reporting data are fine. The introductory figure has some problem with the colors which are not visibly very different and create confusion. Orange and Red do not appear very different in the printed figure.

We have changed this color scheme. 

I do not have major criticisms, but I suggest the authors to rewrite part of the Introduction, is over too long (many details are not necessary), and give more emphasis to the Discussion, which nearly does not exist.

We have shortened the Introductions somewhat, but feel that the remaining details are necessary to convey the proper context of GP action on Ca channels, which is somewhat complex and requires some description of the channels and their accessory subunits. 

 1 - In the Results & Discussion the first sentence: “In following up a recent study examining the role of α2δ subunits in regulating Cav channel expression, we were surprised to find that Gabapentin failed to reduce calcium current density in rat SCG neurons, despite expressing α2δ-1, & 2, and CaV2.2 & 2.3 (Scott & Kammermeir, 2017)” is not correct. The authors did not use and not even mention GP in their paper. The sentence should be rewritten.

We did not intend to suggest that the GP experiments were done IN that study, but that we examined GP actions as a consequence of that study. We have changed the sentence to hopefully clear up this confusion. 

2 - The authors do not discuss the various possibilities by which GP is inactive on the Cav channels of SCG compared to NDG Cav channels. Are the two types of Cav channels structurally similar? do they interact similarly with the a2d subunits? etc.

We have added a section to the new Discussion that addresses some of these possibilities. 

Round 2

Reviewer 1 Report

Comments and Suggestions for Authors

No comments.

Comments on the Quality of English Language

 Minor editing of English language required.